# Radioactive contamination in feral dogs in the Chernobyl exclusion zone: Population body-burden survey and implications for human radiation exposure

Jake Hecla[1,2]*, Erik Kambarian[2], Robert Tubbs[3], Carla McKinley[1], Aaron J. Berliner[1,4], Kayla Russell[5], Gabrielle Spatola[6], Jordan Chertok[7], Weston Braun[8], Natalia Hank[9], Courtney Marquette[5], Jennifer Betz[2], Terry Paik[2], Marie Chenery[2], Alex Cagan[10], Carl Willis[11], Tim Mousseau[6]

1 Department of Nuclear Engineering, University of California Berkeley, Berkeley, California, United States of America, 2 Clean Futures Fund, Godfrey, Illinois, United States of America, 3 Tubbs Nuclear Consulting, Auburn, Washington, United States of America, 4 Department of Bioengineering, University of California Berkeley, Berkeley, California, United States of America, 5 School of Veterinary Medicine, University College Dublin, Dublin, Ireland, 6 Department of Biological Sciences, University of South Carolina, Columbia, South California, United States of America, 7 College of Veterinary Medicine, University of Tennessee, Knoxville, Tennessee, United States of America, 8 Department of Electrical Engineering, Stanford University, Palo Alto, California, United States of America, 9 College of Veterinary Medicine, Kansas State University, Manhattan, Kansas, United States of America, 10 Wellcome Sanger Institute, Hinxton, United Kingdom, 11 Department of Nuclear Engineering, University of New Mexico, Albuquerque, New Mexico, United States of America

* jake_hecla@berkeley.edu

**Data Availability Statement:** All data presented here is available through github: https://github.com/aaronreichmenberliner/dogsofchernobyl.

## Abstract

This report describes a two-year effort to survey the internal $^{137}$Cs and external $\beta$-emitter contamination present in the feral dog population near the Chernobyl nuclear power plant (ChNPP) site, and to understand the potential for human radiation exposure from this contamination. This work was performed as an integral part of the radiation safety and control procedures of an animal welfare oriented trap-neuter-release (TNR) program. The measurement program focused on external contamination surveys using handheld $\beta$-sensitive probes, and internal contamination studies using a simple whole-body counter. Internal $^{137}Cs$ burden was measured non-invasively during post-surgical observation and recovery. External $\beta$ contamination surveys performed during intake showed that 21/288 animals had significant, removable external contamination, though not enough to pose a large hazard for incidental contact. Measurements with the whole-body counter indicated internal $^{137}$Cs body burdens ranging from undetectable (minimum detection level $\sim$100 Bq/kg in 2017, $\sim$30 Bq/kg in 2018) to approximately 30,000 Bq/kg. A total of 33 animals had $^{137}$Cs body-burdens above 1 kBq/kg, though none posed an external exposure hazard. The large variation in the $^{137}Cs$ concentration in these animals is not well-understood, could be due to prey selection, access to human food scraps, or extended residence in highly contaminated areas. The small minority of animals with external contamination may pose a contamination risk allowing exposures in excess of regulatory standards.

**Funding:** This study was funded by the Clean Futures Fund. However, the authors listed under the Clean Futures Fund did not receive any compensation for their time, and no financial transactions were made or distributed to any other co-author institutions. Tubbs Nuclear Consulting provided support in the form of salary for RT. The funders had no role in study design, data collection and analysis, decision to publish, or preparation of the manuscript. The specific roles of these authors are articulated in the 'author contributions' section.

**Competing interests:** RT was employed by the company Tubbs Nuclear Consulting (and likewise volunteered his time). The remaining authors declare that the research was conducted in the absence of any commercial or financial relationships that could be construed as a potential conflict of interest. There are no patents, products in development or marketed products associated with this research to declare. This does not alter our adherence to PLOS ONE policies on sharing data and materials.

## Introduction

On April 26th, 1986, reactor four at Chernobyl nuclear power plant suffered a catastrophic accident, breaching the reactor building and venting approximately 1.3 EBq of radioactive material into the environment over the following weeks [1]. Over thirty years later, the quantities of radioactive material present in the environment in regions of Ukraine, Belarus, and Russia pose a threat to human health [2–4]. Likewise, wild animal populations in these regions maintain high body-burdens of radioisotopes such as $^{137}Cs$ and $^{90}Sr$[5–9]. During the evacuation of population centers near the power plant subsequent to the accident, a large number of domestic animals were released [10]. Despite early efforts at culling the population [11], these animals reproduced rapidly, leading to an enduring population of feral dogs and cats within the exclusion zone. Though highly accurate population estimates are not available, manual survey methods performed by Clean Futures Fund suggest approaching 1000 dogs were present in the immediate vicinity of Chernobyl Nuclear Power Plant in the spring of 2017.

Prior to the invasion by Russian forces in February 2022, the exclusion zone was an active work-site with an electrical switch-yard and extensive cleanup and stabilization activities [12, 13]. Due to the notoriety of the site, there has existed a large tourist presence in the last decade, with visitors numbering in the tens of thousands visiting Pripyat and the exclusion zone every year [14]. While the dogs in the exclusion zone are popular with invited visitors and locals, there are long-running concerns about the risks the feral dog may pose to workers and tourists [15]. Human-animal interaction in the Chernobyl exclusion zone is widespread, and the risks posed by external exposure and contamination transfer have only been studied to a limited extent. Depending on circumstances, the animals may pose a physical threat, biohazard (rabies, in particular), or radiological hazard. While often appearing friendly, the animals at the site are feral, and pose a bite risk to workers and tourists, as well as generating concern about animal welfare.

As a consequence of living in a contaminated environment, these animals may accumulate radioactive dust, move contaminated soil or objects, and transfer contamination to workers or tourists who interact with them [5, 11, 16]. Transfer of external contamination to hands or clothing may create exposure hazards that persist long after a visit to the site. This contamination may be inhaled or ingested, greatly increasing the incurred dose [17]. Site personnel have expressed concerns that highly contaminated animals may even pose a direct, external exposure hazard with prolonged interaction. However, to date no large studies of internal and external radioactive body-burden have been made, and these risks have remained largely unquantified.

In an attempt to stabilize the population and reduce the risk of bites, Clean Futures Fund (CFF, https://www.cleanfutures.org/, along with partners such as the The Society for the Prevention of Cruelty to Animals—International (SPCAi, https://www.spcai.org/) operated a trap-neuter-release (TNR) program in the exclusion zone between 2017 and 2019. To better understand and limit the radiation dose to personnel and workers, monitoring and survey programs were developed to track volunteer dose and to understand the source term associated with the animals. These personnel safety programs had the additional benefit of producing data on animal radioactivity that may be more widely applied to radiation protection outside the clinical setting. These efforts to characterize the transfer of radioisotopes from the animals may be of growing importance as the Chernobyl Exclusion Zone seeks to increase its industrial capacity and develop a solar power station [14, 18]. These activities will dramatically increase the instances of interaction between humans and the feral dogs of Chernobyl.

## Methods

This investigation focused on establishing estimates of external and internal radioactivity of dogs captured in the Chernobyl exclusion zone. This program focused first on rapidly and safely assessing external beta- and gamma-emitter contamination of animals at the entry to the clinic, and later assessing the $^{137}$Cs body burdens of the animals during the recovery phase of the neuter and vaccination process. The measurements performed on the animals were non-invasive and served the immediate purposes of assuring personnel safety and informing future radiation control procedures for similar clinics.

Ethical review and approval for the animal study was conducted with the permission of the Chernobyl Nuclear Power Plant authorities under the supervision of licensed veterinarians andveterinary technicians. A letter of support detailing the cooperation between the CFF and the Chernobyl NPP was signed by the NPP acting general director V.A. Seyda and can be found in the Github outlined below. Approval for access the exclusion zone was handled and granted by the Exclusion Zone Authority. Data collected for this paper were gathered adventitiously while animals were being treated by the medical program and as such are exempt from IACUC approval. Further, the radiation control and measurement activities described in this study were necessary measures for personnel safety during the animal-welfare project.

### External contamination survey

External contamination on animals was assessed at clinic entry to prevent transfer of contamination to "clean" areas of the building. Prior to shaving and intake to surgery, the animals were "frisked" with a Ludlum model 26-2 probe held within 1cm of the fur surface to determine if any external contamination was present [19]. A threshold of 100cpm was used as the minimum count rate (with probe stationary) above which decontamination was needed before dogs could progress through the clinic. Assuming equal distribution over the area below the detector, this threshold is equivalent to a $^{90}$Sr concentration of approximately 1Bq per square centimeter.

Several animals demonstrated persistently elevated count rates on extremities that did not respond to washing or shaving. This was interpreted as internal contamination (likely in bone) due to radioisotopes like $^{90}$Sr. This finding is discussed further in results.

### Whole-body contamination assessment

The whole body counter experiment was intended to quantify $^{137}$Cs body-burden in the captured animals in the post-surgical phase of the clinic. This measurement was taken during the "recovery" period following surgery in which animals must be monitored. The basic arrangement of this system consisted of a shielded, high-efficiency NaI(Tl) detector placed close to the flank of the animal being measured. Gamma-ray spectroscopy was used to identify the $^{137}$Cs photopeak and to subtract background.

The whole-body counter system described in this project used two different detector and shielding arrangements. Due to material transportation constraints, the 2017 phase of the experiment used a thick lead "apron" with rather than a full enclosure as originally intended. The shielding apron consisted of 500 lbs of lead in the form of bricks, which were arranged to create a low-background pad in a corner of the clinic. This consisted of a 2" thick lead apron with a 6" high rim around three sides placed in the lowest-background corner of the clinic, which was then designated the post-surgical recovery area. A frame made of 80-20 extrusion was used to position and hold the cylindrical NaI(Tl) detector at an adjustable height above the animal The first ten or so animals were measured with a volunteer holding the NaI(Tl) detector, rather than using the frame. Measurements were taken by positioning the probe

within 2 cm of the animal's chest with a disposable paper barrier in place to prevent contamination transfer. Gamma rays were counted using a Berkeley Nucleonics SAM940 equipped with a 3" right-cylindrical NaI(Tl) probe, which has a resolution of approximately 8% FWHM at 662keV in this application.

Due to the distribution of contamination in the clinic (mostly on the floor, below plastic sheeting), this arrangement of lead lowered the background in the $^{137}$Cs peak from approximately 50 cps to lower than 18cps with the detector elevated 15cm over the center of the lead apron. Background measurements were re-assessed frequently due to the possibility of contamination, which was mitigated using removable cardboard sheets. Simulation and measurements described in the methods section established a sensitivity threshold of approximately 100Bq/kg for this method of measurement.

In 2018, the system design was revised to improve background rejection and increase sensitivity. This system included an aluminum 80/20 structure to support 1/2in lead shielding on all sides. The enclosure measured 28"W × 28"H × 36" D. The floor of the detector assembly was lined with approximately 500 lbs of lead bricks from the prior system, and a "detector niche" was built to shield the sides and back of the NaI(Tl) cube with 3" of lead (leaving one side open to align with the dog's center of mass). Though the lead wall thickness was limited by the quantity of lead available and the stress limits of the enclosure, the background suppression achieved was significantly improved with respect to the lead apron system. A Harshaw 4" NaI (Tl) cube scintillator read out using a URSA-II 4096-channel MCA was used for data acquisition. In this configuration, peak resolution at 662 keV was approximately 10% FWHM. The background rate measured with this system was approximately 15 cps in the Cs peak (in comparison with an unshielded background of approximately 150 cps), and boasts a factor of 3 improvement in sensitivity over the 2017 system.

The lack of graded-Z shielding in both system designs limited the ability of the system to see low-energy features such as the 59 keV emission line from $^{241}$Am and the numerous low-energy lines from $^{239}$Pu which may have been present. The high-energy $^{137}$Cs line, was easily observed due to the optical-thinness of the dogs and the high detection efficiency of NaI(Tl) at that energy. As a result, this study was limited to Cs, though future investigations using improved shielding and higher resolution detectors could expand this work to other isotopes of interest.

## Monte Carlo simulations and calibration

Monte Carlo photon transport calculations were performed to determine the correlation between photopeak counts and 137[Cs] body burden for the detector and shielding designs described here. Though methods exist to use the equilibrium concentration of $^{40}$K as an *in vivo* calibration, the abundance of K in the surrounding concrete in the clinic made this method prone to error. To provide a consistent calibration for animals of any weight, a model was created using measurements of the torsos of four animals from the cohort. This model was then used to create input geometries representing animals weighing between 1 and 35 kg.

These torso models used International Commission on Radiation Protection (ICRP) specifications for cortical bone and skeletal muscle tissue. Though some portions of the anatomy were removed for simplicity, the mean free path of 662 keV gamma rays in tissue is significantly longer than the characteristic dimensions of any animal surveyed, rendering these changes unlikely to contribute significantly to error. Two different input templates were developed for each whole body counter version. These simulations provide a calibration allowing count-rates to be related to $^{137}$Cs body burdens without use of a phantom or the $^{40}$K ratio comparison, which is subject to extreme variation due to building materials near the detector.

## Data processing

Gamma-ray spectra from each dog were recorded, along with information on animal ID number, tag number, weight and comments on decontamination status or significant medical conditions. Background measurements were performed frequently, and noted in a laboratory notebook to compensate for any contamination entering or leaving the clinic area.

A MATLAB script was used to read the spectra files, and select appropriate background files for each. For each animal, counts in the $^{137}$Cs peak were summed and subtracted from the background rate for that time interval. The animals' weights were then used to select an interpolated counts-to-body-burden conversion factor from the simulation outputs. The excess counts in the peak region were then used to determine the body burden estimate. Counting error, as well as error from weight measurements and interpolation uncertainty were included in the calculated uncertainty in the body burden.

## Results

A total cohort of 252 animals were captured and treated in the clinic in 2017, and 36 were treated in 2018 (Fig 1). We note that in order to minimize variability, only measurements

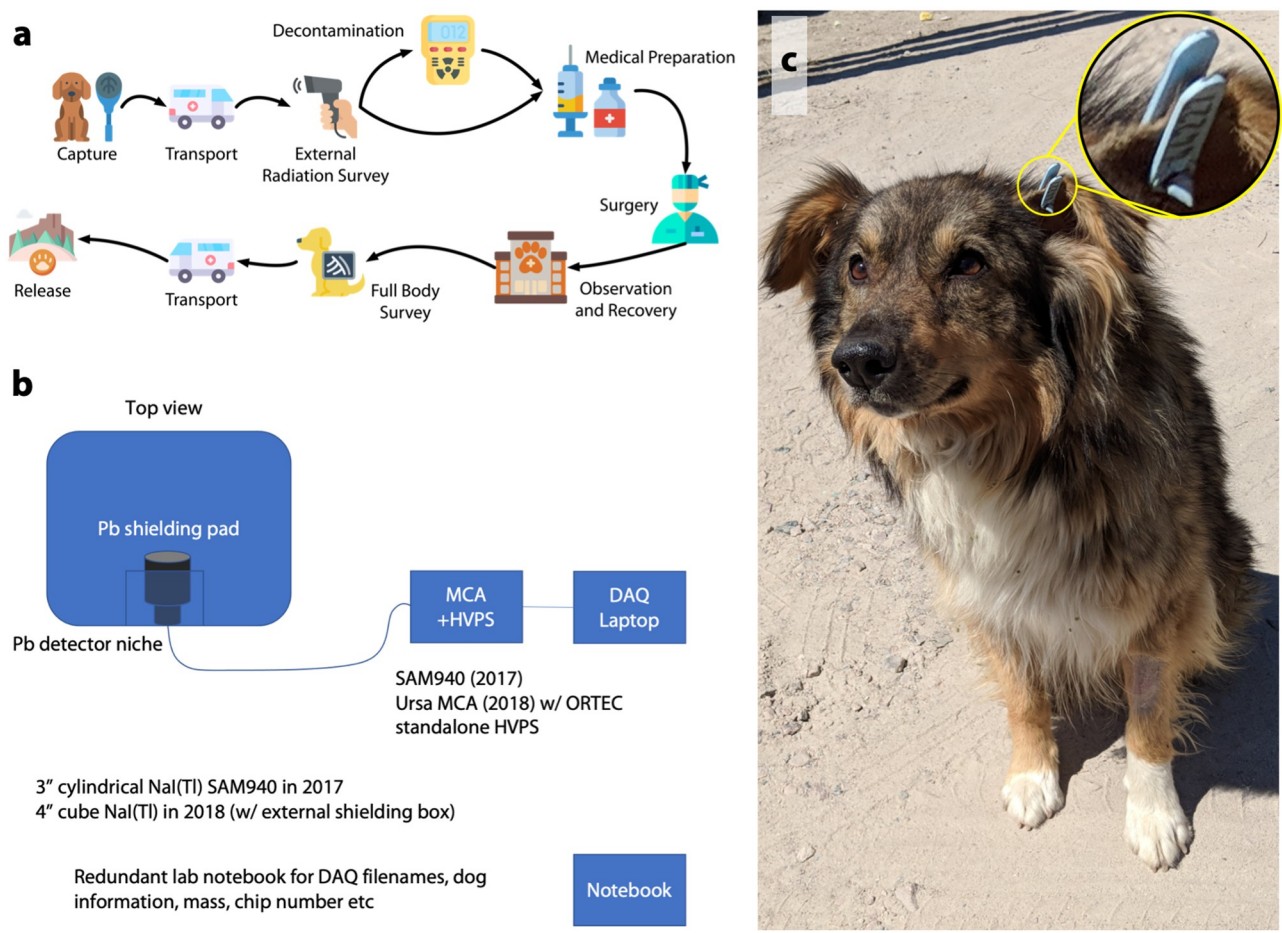

**Fig 1. Methods and photos.** (a) Diagram of experimental approach. (b) Diagram of experimental hardware and systems. Native dogs were collected and initially triaged before being measured. Numbered, thermoluminescent dosimeter-equipped ear tags (c) were place on dogs during the TNR campaign as part of a related study.

taken under the supervision of the lead researcher are included in this study. Due to the limited presence of the lead researcher, the 2018 data set encompasses only the measurements taken on the first two days of the 2018 TNR campaign.

All animals entering the clinic were surveyed using a handheld $\beta/\gamma$ sensitive Ludlum 26-2 "pancake" probe with removable alpha shield to assess external contamination hazards before clinic entry. A total of 198 animals randomly selected from the cohort of 288 were subject to observation using the whole-body counter. Captured animals ranged in mass from 0.9 kg to 35 kg, with a sex ratio of 7:3 biased towards females. We also note that animal capture technicians in the field carried pen-type (RADEX One) active dosimeters with alarm features to provide warning should they encounter highly contaminated animals that could provide an immediate radiation threat, though this was never observed in practice.

External contamination assessed with the pancake probe above threshold (defined as 100 cpm, approximately 95 Bq per 100 cm$^2$, assuming 22% efficiency for $^{90}$Sr) was observed on 21 of 288 animals, primarily on paws and forelegs. Assuming $^{90}$Sr as the dominant beta emitter (which is true in most areas of the zone), contamination levels ranged up to 1.9 kBq per 100 cm$^2$. We note that averages with a areal value over 100cm$^2$ have been made for all external contamination measurements to prevent misleadingly high estimates due to highly localized contamination.

All animals with detectable external contamination were decontaminated by washing and/ or shaving until surveys showed contamination below the 100cpm threshold. No clear trends based on sex, body mass, or accompanying medical conditions were apparent in either internal or external contamination measurements performed. Several animals initially believed to be externally contaminated which did not respond to decontamination efforts were hypothesized to have an internal $\beta$-emitter body-burden on bone surfaces which was externally detectable. This was likely due to $^{90}$Sr deposited on bone surfaces, though no further investigation was made in this study. Note that these animals were included in the whole-body counter measurements, though none of them showed $^{137}$Cs burdens above the minimum detection threshold.

The external $\beta$-emitter contamination discovered on animals was found to be highly transferable, with towels, instruments, and cleanup implements requiring frequent washing due to contamination from contact with the dogs prior to decontamination. One sling used for animal transfer was found to be contaminated to approximately 900 Bq per 100 cm$^2$ (assuming $^{90}$Sr) over most of its surface. Volunteers involved in the intake process were frequently subjected to external contamination assessments, and were occasionally found to have contamination transfer ed from animals on clothing or gloves.

Of the 198 animals examined using the whole-body counter for internal contamination, 91 were found to have measurable $^{137}$Cs contamination ranging from 30-30,000 Bq/kg. None of the spectra gathered showed clear evidence of other radioisotopes aside from $^{137}$Cs. Most animals with measured external $\beta$ contamination also had measurable internal $^{137}$Cs body-burdens. This body-burden signal is unlikely to be due to remaining external contamination, as the animals had to be fully decontaminated and surveyed before entering the operating room and recovery areas. We note that external $^{137}$Cs contamination results in $\beta$-emissions which are detected with similar efficiency to $^{90}$Sr using the Ludlum 26-2 probe.

In 2017, with a limit of sensitivity of approximately 100 Bq/kg, 62/81 animals captured within the power plant boundaries were detectably internally contaminated, versus 1/82 in the exclusion zone. In 2018, with a lower sensitivity threshold of approximately 30 Bq/kg, 20/24 animals from the power plant were detectably internally contaminated, whereas 8/11 from the exclusion zone had measurable $^{137}$Cs.

The same trend was evident in external contamination. In 2017, 15/114 animals captured within the power plant boundaries were externally contaminated, in contrast with 1/138 of

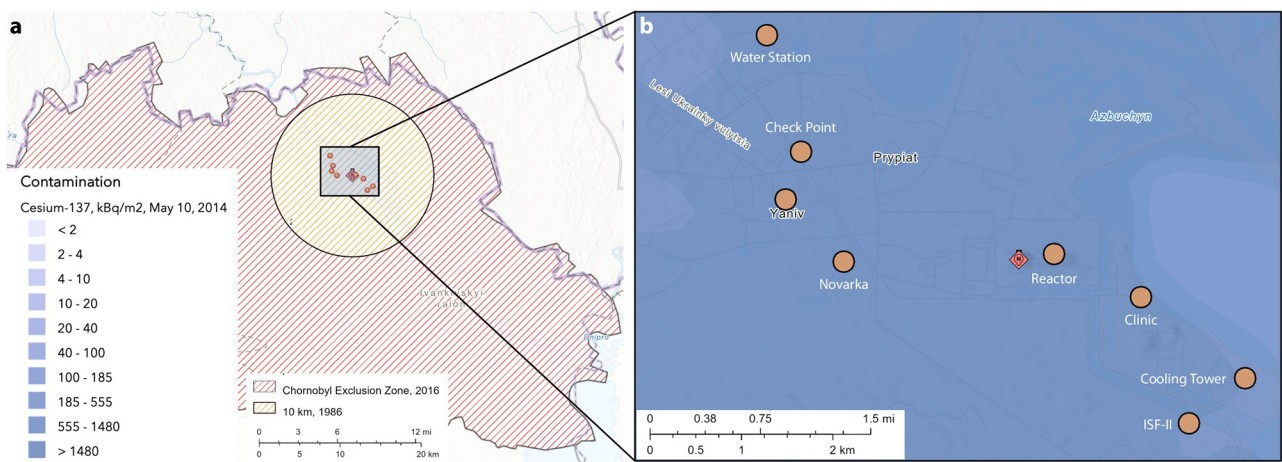

**Fig 2. Understanding the Chernobyl region of interest. (a)** Map of Chernobyl exclusion zone (2016) and 10 km radius inner circle. **(b)** Exploded view of points of interest (orange circles) within 10 km radius inner circle centered on the ChNPP (red diamond), colored by $^{137}Cs$ contamination (2014). (Access: https://harvard-cga.maps.arcgis.com/home/index.html).

those outside the boundary. In 2018, 4/24 animals captured in the power plant were externally contaminated, whereas only 1/12 from the exclusion zone tested above threshold.

The most internally contaminated animals captured in the 2017 campaign were captured near the ISF-II fuel storage facility ("ISF-II" in Fig 2b). The 2018 dataset showed that the most internally contaminated animals were captured in the reactor Local Zone and near the Novarka site, both locales extremely proximate to the reactor itself ("Reactor" and "Novarka" in Fig 2b).

## Discussion

Internal and external radioisotope contamination has been observed in animals taken from both the power plant boundary and exclusion zone, with animals in the former displaying significantly higher mean levels of internal and external radionuclide contamination. No animals were measured with levels of internal contamination that would pose an external exposure hazard in passing. Contact with the most internally contaminated animals would result in an exposure rate several times that of background in the zone (assuming travel along major tourist routes), but would not violate guidelines on maximum dose rate for members of the general public (for reference, the US public is not to be exposed to more than $20\mu$ Sv/hr). The observed external doserates are negligible in comparison with other risks posed by working in or visiting the exclusion zone.

Dogs with removable external contamination, however, were observed and are likely to pose a small internal contamination risk to humans resulting from transfer of radioactive material to skin and clothing when interacting with the dogs. This transferred contamination could easily be inhaled or ingested if proper decontamination procedures are not followed. In several cases, significant detectable activity was transferred to clothing and equipment in the "dirty" portion of the clinic, confirming that such transfer is possible.

We conclude that internally contaminated dogs are unlikely to pose an external exposure risk to persons interacting with them based on the dose rates measured in this cohort. The highest body-burden dog in this study measured approximately 1 $\mu$Sv/hr on contact (after external contamination was removed, as measured with a SAM-940 equipped with a 3" NaI

**Table 1. Instances of internal activity detection as determined by the whole-body counting experiment.** Note that the 2017 experiment had a detection threshold of approximately 100 Bq/kg, whereas the 2018 revision was sensitive to approximately 30 Bq/kg.

| Location | No Detectable $^{137}$Cs | Detectable $^{137}$Cs | Total |
|---|---|---|---|
| ChNPP, 2017 | 19 | 62 | 81 |
| Exclusion Zone, 2017 | 81 | 1 | 82 |
| ChNPP, 2018 | 4 | 20 | 24 |
| Exclusion Zone, 2018 | 3 | 8 | 11 |
| *ChNPP Total* | 23 | 82 | 105 |
| *Exclusion Zone Total* | 84 | 9 | 93 |
| **Cohort Total** | 107 | 91 | 198 |

(Tl)) detector. While this significantly exceeds the background dose rate in much of the zone, it is very small in absolute terms. We note that patients administered internal gamma emitters in a medical setting are not to cause exposure exceeding 20 $\mu$Sv in any hour on discharge, which is some 20× higher than the most active dog in the cohort [20].

In contrast to dogs with high internal body-burdens, dogs with removable external contamination may pose a handling hazard. The dominant $\beta$-emitting radioisotope at Chernobyl is $^{90}$Sr, which is a highly bioavailable calcium analogue. External contamination levels up to 1.9 kBq per 100 cm$^2$ were observed on a cloth sling used for animal transportation. The contamination levels observed from transfer due to contact with the dogs may result in doses significantly exceeding radiation protection standards if inhaled or ingested.

While exact contamination transfer calculations are not possible in this scenario, the example of the contamination transfer to the transport sling is instructive in calculating approximate doses. Should an individual experience the same level of contamination transfer to clothing observed in the case of the sling, and go on to ingest the amount of $^{90}$Sr present in 10 cm$^2$ of material (perhaps by wiping the corner of their mouth on a sleeve or failing to wash their hands before eating), the dose to the bone surface would be approximately 2.5 mRem (0.22 mRem committed effective dose equivalent (CEDE)). If inhaled, this quantity would result in 17.5 mRem to the lung (2.07 mRem CEDE).

Over the course of this study, most dogs from the ChNPP site (82/105) had measurable $^{137}$Cs burdens, with four dogs exceeding 10 kBq/kg. The proportion of dogs in the exclusion zone (Fig 1) which were internally contaminated was dramatically lower (9/93), and generally had body-burdens near the lower limit of detection. Due to the varying sensitivity of the WBC systems, these ratios are best broken down by year of capture (Table 1). Extensive access to clean food provided by the CFF feeding program may have contributed to an observed decrease in internal contamination levels between 2017 and 2018, though the small sample size from the 2018 campaign limits the significance of this finding.

Externally contaminated dogs were significantly more common on the ChNPP site than in the exclusion zone (19/138 vs 2/150). Rates of external contamination were not significantly different between 2017 and 2018 for either the Exclusion Zone or the ChNPP site. The sites of contamination observed were primarily on paw surfaces and between pads, though a number of dogs had extensive contamination along the entire ventral surface (Fig 3a).

## Conclusion

Feral dogs captured in the Chernobyl Exclusion Zone and power plant premises have detectable internal and external radioisotope body-burdens which may pose a a minor radiological

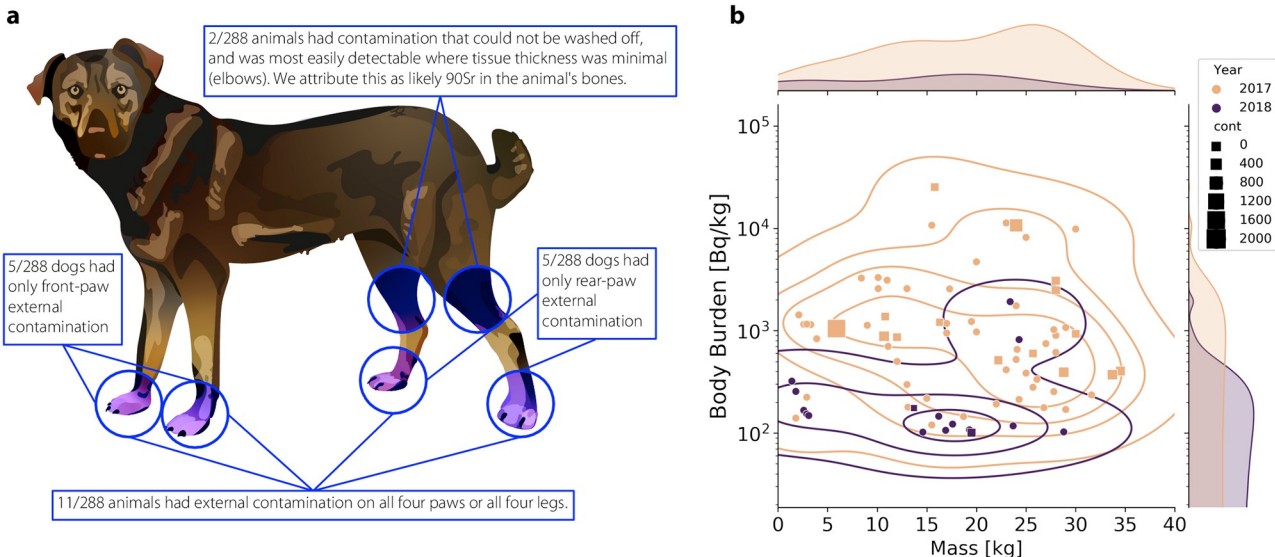

**Fig 3. Dosimetry and analysis of wild dogs. (a)** 21/288 animals surveyed as part of the 2017 and 2018 programs had external contamination above the threshold (100 cpm on Ludlum 26-2, equivalent to 95 Bq per 100 cm$^2$ area, $^{90}$Sr). The distribution of this contamination is shown here. **(b)** Scatter plot of the animal body-burden as a function of body mass for the 2017 (peach) and 2018 (purple) campaign. Animals in 2017 with activity less than 100 Bq/kg are excluded. Animals in 2018 with activity less than 30 Bq/kg are excluded. External contamination is shown in this plot using squares, the size of which is correlated to the reported measurement in counts per minute using the Ludlum 26-2 probe on contact.

hazard to those interacting with the animals. The magnitude of the internal body-burden detected in the dogs ranged from undetectable to approximately 30 kBq/kg. External contamination levels ranged from undetectable to approximately 20 Bq/cm$^2\beta$. While the internal $^{137}$Cs body-burden detected in the dogs did not in any case represent an exposure hazard to those handling the animals, the external contamination detected on the dogs' fur surveyed in this study represents a potential source term for human exposure if proper radiological hygiene practices are not followed. We assess that individuals coming into contact with the animals and practicing poor radiological hygiene could be exposed to doses in excess of limits for the general public. We further determine that the risk of exposure depends on both the extent of interaction and the location in which the animals typically reside. Animals from areas near the reactor site are more likely to be contaminated, and are more likely to carry a higher radioisotope burden, as discussed. These trends hold true for both internal and external contamination.

Going forward, those coordinating and supervising work in the Chernobyl exclusion zone and power plant area should assure that workers and tourists in the areas nearest the power plant are properly cautioned on the additional unseen risks of interacting with the feral animals [21]. Future work should focus on improving the accuracy and portability of the whole-body counter system used in this investigation, as well as selecting more sensitive $\beta$-counting instrumentation.

## Acknowledgments

First and foremost, we thank the administrators and staff of Chernobyl Nuclear Powerplant (SSE-ChNPP), the Chernobyl Exclusion Zone authority, the city of Slavutych and the Novarka consortium. In particular, Stanislav Shekstelo was of great assistance in this effort. Sincerest thanks for the technical support, encouragement, general assistance, and lead shielding

supplies provided by Dr. Tim Mousseau's laboratory and collaborators in Chernobyl town during the course of this project. Finally, we thank Davian Ho for his help with the graphics.

## Author Contributions

**Conceptualization:** Jake Hecla, Tim Mousseau.

**Data curation:** Jake Hecla, Carla McKinley, Carl Willis.

**Formal analysis:** Jake Hecla, Robert Tubbs, Carla McKinley, Aaron J. Berliner, Carl Willis, Tim Mousseau.

**Funding acquisition:** Jake Hecla, Tim Mousseau.

**Investigation:** Jake Hecla, Erik Kambarian, Kayla Russell, Gabrielle Spatola, Jordan Chertok, Weston Braun, Natalia Hank, Courtney Marquette, Jennifer Betz, Terry Paik, Marie Chenery, Alex Cagan, Carl Willis, Tim Mousseau.

**Methodology:** Jake Hecla, Carl Willis.

**Project administration:** Jake Hecla, Aaron J. Berliner, Tim Mousseau.

**Resources:** Jake Hecla, Robert Tubbs, Carl Willis.

**Software:** Robert Tubbs.

**Writing – original draft:** Jake Hecla, Carla McKinley.

**Writing – review & editing:** Jake Hecla, Carla McKinley, Aaron J. Berliner, Kayla Russell, Gabrielle Spatola, Jordan Chertok, Weston Braun, Natalia Hank, Courtney Marquette, Jennifer Betz, Terry Paik, Marie Chenery, Alex Cagan, Carl Willis, Tim Mousseau.

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
