## [Decision Letter · Decision Letter 0]

27 Sep 2022

PONE-D-22-14715Radioactive Contamination in Feral Dogs in the Chernobyl Exclusion Zone: Population Body-Burden Survey and Implications for Human Radiation ExposurePLOS ONE

Dear Dr. Berliner,

Thank you for submitting your manuscript to PLOS ONE. After careful consideration, we feel that it has merit but does not fully meet PLOS ONE’s publication criteria as it currently stands. Therefore, we invite you to submit a revised version of the manuscript that addresses the points raised during the review process.

On reviewer positively noted: “Thank you for the opportunity to review this interesting manuscript for PLOS One. The conclusions drawn may have a big impact for the radiation protection of radiation workers at the Chernobyl site as well as the public.”

Again, our apologies for the delay and the need to send the manuscript out several times for re-review due to unavailability of qualified reviewers and some initial disparities between positive and negative reviews. I am pleased that the reviews now indicate only minor revision is needed.

We look forward to receiving your revised manuscript.

Kind regards,

Norman J Kleiman, PhD

Academic Editor

PLOS ONE

Journal Requirements:

5. We note that Figure 3a in your submission contain copyrighted images. All PLOS content is published under the Creative Commons Attribution License (CC BY 4.0), which means that the manuscript, images, and Supporting Information files will be freely available online, and any third party is permitted to access, download, copy, distribute, and use these materials in any way, even commercially, with proper attribution. For more information, see our copyright guidelines: http://journals.plos.org/plosone/s/licenses-and-copyright.

    a. You may seek permission from the original copyright holder of Figure 3a to publish the content specifically under the CC BY 4.0 license.

Additional Editor Comments :

Dr. Berliner,

Apologies for the delay in returning your manuscript. It was difficult finding bandwidth from qualified reviewers over the summer. Initially, there was some disagreement between reviews and so we subsequently asked for additional eyes on the manuscript. I am pleased to report that the majority of reviews now ask for minor revisions. We look forward to receiving a revised manuscript.

Reviewers' comments:

Reviewer's Responses to Questions

**Comments to the Author**

1. Is the manuscript technically sound, and do the data support the conclusions?

Reviewer #1: Yes

Reviewer #2: No

Reviewer #3: Partly

2. Has the statistical analysis been performed appropriately and rigorously? 

Reviewer #1: No

Reviewer #2: N/A

Reviewer #3: N/A

3. Have the authors made all data underlying the findings in their manuscript fully available?

Reviewer #1: Yes

Reviewer #2: No

Reviewer #3: Yes

4. Is the manuscript presented in an intelligible fashion and written in standard English?

Reviewer #1: Yes

Reviewer #2: Yes

Reviewer #3: Yes

5. Review Comments to the Author

Reviewer #1: The paper from Hecla J. et al. aims to assess internal and external contamination of feral dogs at Chernobyl and possible contamination transfer to humans.

The main conclusions are that internal contamination due to accumulation of 90Sr in bones would not result in external contamination to humans interacting with the dogs for a short period of time. In contrast, dogs with removable external contamination may pose health risks to humans interacting with them caused by contamination transfer.

Here are few comments and some curiosity-driven questions:

- During neutering surgery, were gross tumor-like growths observed in any part of the body? Did some of the dogs look ‘’sick’’? My question relates to possible radiation-induced soft tissues cancers and/or symptomatic manifestations of osteosarcomas due to internal contamination.

- It is stated that ‘’Several animals initially believed to be externally contaminated which did not respond to decontamination efforts were hypothesized to have an internal β-emitter body-burden on bone surfaces which was externally detectable. This was likely due to 90Sr deposited on bone surfaces…’’. However, in one of the boxes of Fig 3a it is reported that 1/288 dogs had contamination that could not be washed off. What is the real number?

- It is not clear the information Fig 2b conveys. Please clarify or simply remove it.

- Fig 2c: Was the ear tag dosimeter worn only during the TNR procedure? If not, I wonder if the dogs wouldn’t scratch their ears, and some would do so with their contaminated paws and maybe affect the dosimeter reading.

- Paraphrasing from the discussion, internal contamination would not pose a health risk to ‘’passersby’’ who would not spend hours interacting with contaminated animals (and therefore exceed the limit of 20 uSv/h). However, what about veterinarians and staff of the TNR program? How long does any procedure take and how many procedures they handle a day?

-Basic statistical analysis should be provided, as for example to support the statement that internal contamination level correlated with capture location.

Reviewer #2: Disaster area affected by the nuclear disaster, TNR activities are important due to the overgrowth of animal population and ethical considerations. The authors analyzed internal and surface contamination of dogs protected by TNR and evaluated exposure levels of radioactivity. There are few reports of these activities internationally, and it is useful to disseminate information analyzed by academic organizations internationally. It is worth noting that this study evaluated the risk of contamination by towels and the like contaminated during dog care and the risk to workers, tourists, and other people interacting with the dog. However, unfortunately, detailed data was not shown in the paper, and the details of the data could not be confirmed at the database address shown. In order for this study to provide scientific information to interested readers, it is recommended that the paper provide details of contamination level distribution in the dog populations. Furthermore, from the viewpoint of animal ethics, it is desirable to comprehensively provide information on the health status and infectious diseases of protected dogs. I recommend that author’s activities be shared with readers, along with detailed data, statistically analyzed results, and individual health information.

Reviewer #3: Overall an interesting paper with information that has not previously been reported. However, the paper would benefit from calculations of potential dose to persons handling the dogs - as presented here, it is difficult to see that this would pose an unacceptably high risk. In addition, a photo of the dog-counting setup would be helpful.

6. PLOS authors have the option to publish the peer review history of their article (what does this mean?). If published, this will include your full peer review and any attached files.

Reviewer #1: No

Reviewer #2: No

Reviewer #3: No

---

## [Author Response · Author response to Decision Letter 0]

5 Dec 2022

Page

line

Comment

Abstract 

Even high levels of contamination rarely pose a health risk, even to those who are contaminated. Did the authors quantify the amount of contamination present on any of the dogs, calculate the amount that could reasonably transfer to a person, and determine the potential health effects of this transfer? Barring some sort of calculation to this effect, I have to confess to being somewhat dubious about the potential for external contamination on a dog to transfer to a person in sufficient quantity to pose a health risk.

- The quantification of the total surface activity is in the results section, and has been amended to reflect uncertainty in transfer coefficients . While the dose caused by ingesting or inhaling this quantity of material is not large in absolute terms, it is significant above exposure thresholds in place for members of the public. We have edited the paper to reflect this

3

30

Again, I am somewhat dubious about contamination posing a health risk, even if transferred to skin. Can the authors provide some calculations based on average and maximum removeable contamination levels they found, indicating a plausible dose to the skin from incidental contact with contaminated animals, and to the whole body in the event that some of this contamination were to transfer to food and be ingested? I understand that, according to ALARA, any exposure is to be avoided if possible – but I’m also guessing that tourists and animals are likely to be unfamiliar with the ALARA principle , so such an assessment seems reasonable.

-We have added this analysis to the results section. We appreciate these edits, as they significantly improve the paper. 

4

34

Similarly, can the authors calculate exposure to people at a distance of, say, 30 cm and 1 m from the contaminated animals they measured? How does the calculated dose rate compare to the actual dose rates measured?

-Due to the fact the emitters (the dogs) have a spatial extent comparable to the standoff distance, simple inverse square scaling does not work. As a result, we would have to run detailed particle transport calculations to answer this question properly. As the doserates are negligible for external exposure, we have modified the language to reflect the minimal nature of the hazard.

4

75

How does the background noted here compare to unshielded background count rate outside at this location?

-This has been added in. It’s about a factor of ten reduction.

5

96

Do the authors have a citation for the high counting efficiency noted here? It’s not that I doubt them – just that Knoll notes an intrinsic counting efficiency of a little less than 90% for a 4” NaI detector and, when we add to that a geometric counting efficiency of 50% or less, the absolute counting efficiency is more likely around 40-45% for Cs-137.

-This was phrased poorly. We were referring to the intrinsic efficiency, not the overall counting efficiency. This has been revised to clarify.

5

100

Good call on not using K-40! The amount in concrete (and brick, for that matter) depends strongly on the clay mineralogy of these materials

-Thank you. We made this call after attempting this in the lab– the corrections just didn’t work well.

6

131

Can the authors provide a reference (or calculation) for 22% counting efficiency for Sr-90? Also, is this for Sr-90 alone, or are you assuming that Y-90 is in equilibrium? If Y-90 is present, the actual counting efficiency might be much higher due to the presence of the second beta and its much higher energy.

-This is from the user’s manual for the Ludlum 26-2, and assumes the Y90 is in equilibrium. You are correct to point out this, and we have corrected our values to reflect an equilibrium Sr90/Y90 concentration (https://ludlums.com/images/data_sheets/M26-2.pdf) We have also included this reference in the manuscript. 

7

160

How did the authors differentiate between the beta and gamma radiation when determining the counting efficiency? Or is the gamma counting efficiency simply too low to be a factor (in which case, perhaps this can be noted)?

-Gamma efficiency is extremely low (<1%) for the sensor used. Further, a crude gamma/beta discrimination test (flipping the pancake to irradiate the tube from the back, which is shielded by 5mm Fe) was used on all contaminated animals to determine if significant gamma emitter contamination was present.

7

168

Here, too, calculated exposure rates would be helpful. We have addressed this in the text. Thank you for pointing this out.

8

190

Do the authors recommend that anyone coming into physical contact with animals wear gloves, wash their hands, shower, change clothing, etc afterwards to reduce their exposure?

We have added this recommendation.

Can the authors suggest mechanisms by which touching a contaminated animal can result in significant internal contamination and likely exposures? I can think of a few (e.g. eating a sandwich before washing one’s hands) – but this seems unlikely to transfer sufficient contamination to produce a high internal exposure. 

We noted multiple incidents of transfer to the hands and clothes of workers in the clinic requiring decontamination. Notes from the 2017 campaign indicate at least a half-dozen incidents of volunteers requiring clothes be changed or otherwise decontaminated due to contact with the dogs.

8

200

The paper mentions using an “energy-compensated NaI(Tl) detector. I am familiar with using energy-compensated GM detectors and have used many of them over the years. But I have never heard the term applied to sodium iodide. Can the authors indicate which of the NaI(Tl) detectors mentioned was energy-compensated?

-This was an error, thank you for pointing this out. The doserate function in the SAM940 uses the gamma ray energy to determine doserate. However, this is not true “energy compensation,” and this has been removed.

8

201

Thanks for noting that 1 µSv/hr is not a very high dose rate – this might be worth mentioning in the abstract as well, even if only qualitatively.

-We have added additional language to contextualize the doserate. Thank you for pointing this out!

9

217-218

I have to admit that this level of exposure doesn’t seem as though it’s anything to be worried about. I know…ALARA…but it’s also less than I recorded during a flight from NYC to LAX or from Detroit to Amsterdam.

-That is correct that it’s low, but it’s above standards for exposure to the public, hence our concern. We have modified our language a bit to provide additional context.

9

237

I’m not sure that I agree with the word “hazard” given the low doss reported in this paper. ICRP notes a dose of 2-3 mrem as being a “trivial” dose of radiation and most of the doses reported here are in that range. But even a dose of, say, 1 rem gives only 0.05% risk of developing a fatal cancer – compared to the risk from driving or the background cancer mortality rate, even 1 rem does not seem like much of a “hazard.” Is there another word that can be used to indicate that there might be a risk – but that it’s very, very low?

-We have amended this to state that the risk is low, but in excess of regulatory standards. This better reflects the nature of the risk, as you are correct in that “hazard” conveys too strong of a risk.

Can the authors include a photograph (or at least a detailed drawing) of their canine whole-body counter (preferably with a canine being counted)? 

-Due to a camera containing most photos being stolen, we have a fairly limited array of photos showing the system. We have added a figure that shows the device in action, along with a head-on photo of the device. However, the head-on photo of the system unfortunately was taken when one researcher was measuring his own Cs body burden in the system. His backside has therefore been edited out of the photo, which will be mentioned in the caption.

- During neutering surgery, were gross tumor-like growths observed in any part of the body? Did some of the dogs look ‘’sick’’? My question relates to possible radiation-induced soft tissues cancers and/or symptomatic manifestations of osteosarcomas due to internal contamination.

-The animals did not appear “sick” in the overwhelming majority of cases, though the average age of the animals was noted to be rather low. To quote Dr. Betz: “Nothing is noted. However, must take into consideration that it is a very small incision and we reach in with a hook and pull the uterus out so we are not physically exploring the abdomen. There is a possibility of tumors in there, however, highly unlikely at that age.”

- It is stated that ‘’Several animals initially believed to be externally contaminated which did not respond to decontamination efforts were hypothesized to have an internal β-emitter body-burden on bone surfaces which was externally detectable. This was likely due to 90Sr deposited on bone surfaces…’’. However, in one of the boxes of Fig 3a it is reported that 1/288 dogs had contamination that could not be washed off. What is the real number?

Thank you for pointing this out, the number has been revised. Two animals (dogs O007 and O068) were contaminated. Figure 3 has been updated in the manuscript. 

- It is not clear the information Fig 2b conveys. Please clarify or simply remove it.

Thank you for pointing this out. We have updated Figure 2 and removed the original Figure 2b.

- Fig 2c: Was the ear tag dosimeter worn only during the TNR procedure? If not, I wonder if the dogs wouldn’t scratch their ears, and some would do so with their contaminated paws and maybe affect the dosimeter reading.

The details about the ear tag study have been removed from the discussion. You are correct that this is a potential confounding factor, but this study is outside the scope of this paper. 

- Paraphrasing from the discussion, internal contamination would not pose a health risk to ‘’passersby’’ who would not spend hours interacting with contaminated animals (and therefore exceed the limit of 20 uSv/h). However, what about veterinarians and staff of the TNR program? How long does any procedure take and how many procedures they handle a day?

-Procedures took between 5 and 20min. Depending on dog capture rate and clinic operations, each vet would perform between 10 and 40 procedures per day. Personal dosimeters (RADEX One) were worn by all clinic staff, and total doses did not not exceed 10uSv for any individual for any day. The highest accumulated doses were encountered by staff working in the field, rather than those in the clinic.

No animals surveyed had an external doserate on contact above 1uSv/hr. As a result, the doserates at 30cm are significantly lower than that. I cannot provide doserates at 30cm and 1m without resorting to detailed modeling, as the emitter dimension is of a similar scale to the standoff distance. 

-Basic statistical analysis should be provided, as for example to support the statement that internal contamination level correlated with capture location.

Thank you for pointing this out. We have updated the manuscript and removed the problematic statement. 

Reviewer #2: Disaster area affected by the nuclear disaster, TNR activities are important due to the overgrowth of animal population and ethical considerations. The authors analyzed internal and surface contamination of dogs protected by TNR and evaluated exposure levels of radioactivity. There are few reports of these activities internationally, and it is useful to disseminate information analyzed by academic organizations internationally. It is worth noting that this study evaluated the risk of contamination by towels and the like contaminated during dog care and the risk to workers, tourists, and other people interacting with the dog. However, unfortunately, detailed data was not shown in the paper, and the details of the data could not be confirmed at the database address shown. In order for this study to provide scientific information to interested readers, it is recommended that the paper provide details of contamination level distribution in the dog populations. Furthermore, from the viewpoint of animal ethics, it is desirable to comprehensively provide information on the health status and infectious diseases of protected dogs. I recommend that author’s activities be shared with readers, along with detailed data, statistically analyzed results, and individual health information.

We thank the reviewer for this comment. We apologize for not uploading our materials to the referenced github sooner. All materials have been uploaded to the github including

A compiled dataset (“data1.xlsx”)

Jupyter notebook for plotting

Figure components and construction files

Photos

Reviewer #3: Overall an interesting paper with information that has not previously been reported. However, the paper would benefit from calculations of potential dose to persons handling the dogs - as presented here, it is difficult to see that this would pose an unacceptably high risk. In addition, a photo of the dog-counting setup would be helpful.

We thank the reviewer for this comment. We have updated the manuscript where possible and added more photos in Figure 2 of the dog-counting setup.

---

## [Decision Letter · Decision Letter 1]

6 Mar 2023

Radioactive Contamination in Feral Dogs in the Chernobyl Exclusion Zone: Population Body-Burden Survey and Implications for Human Radiation Exposure

PONE-D-22-14715R1

Dear Dr. Berliner,

We’re pleased to inform you that your manuscript has been judged scientifically suitable for publication and will be formally accepted for publication once it meets all outstanding technical requirements.

Kind regards,

Norman J Kleiman, PhD

Academic Editor

PLOS ONE

Additional Editor Comments (optional):

Reviewers' comments:

Reviewer's Responses to Questions

**Comments to the Author**

1. If the authors have adequately addressed your comments raised in a previous round of review and you feel that this manuscript is now acceptable for publication, you may indicate that here to bypass the “Comments to the Author” section, enter your conflict of interest statement in the “Confidential to Editor” section, and submit your "Accept" recommendation.

Reviewer #2: All comments have been addressed

Reviewer #3: All comments have been addressed

2. Is the manuscript technically sound, and do the data support the conclusions?

Reviewer #2: Yes

Reviewer #3: Yes

3. Has the statistical analysis been performed appropriately and rigorously? 

Reviewer #2: N/A

Reviewer #3: N/A

4. Have the authors made all data underlying the findings in their manuscript fully available?

Reviewer #2: Yes

Reviewer #3: Yes

5. Is the manuscript presented in an intelligible fashion and written in standard English?

Reviewer #2: (No Response)

Reviewer #3: Yes

6. Review Comments to the Author

Reviewer #2: The revised manuscript entitled “Radioactive Contamination in Feral Dogs in the Chernobyl Exclusion Zone: Population Body-Burden Survey and Implications for Human Radiation Exposure (PONE-D-22-14715_R1)”, submitted by Jake Hecla et al. reported radioactive contamination in feral dogs in the Chernobyl exclusion zone and hazard to workers, tourists, and others interacting with the dogs. The health risks from internally contaminated dogs are extremely low and do not pose a public health concern. However, scientifically clarifying the impact on support staff through the activities of the authors is an important effort to deny reputational damage and excessive anxiety.

Reviewer #3: I appreciate the authors taking the time and making the effort to address my comments. Interesting paper!

7. PLOS authors have the option to publish the peer review history of their article (what does this mean?). If published, this will include your full peer review and any attached files.

Reviewer #2: **Yes: **Tomisato Miura

Reviewer #3: **Yes: **Andrew Karam

---

## [Editor Report · Acceptance letter]

30 Mar 2023

PONE-D-22-14715R1 

Radioactive contamination in feral dogs in the Chernobyl Exclusion Zone: Population body-burden survey and implications for human radiation exposure 

Dear Dr. Berliner:

I'm pleased to inform you that your manuscript has been deemed suitable for publication in PLOS ONE. Congratulations! Your manuscript is now with our production department. 

Kind regards, 

on behalf of

Dr. Norman J Kleiman 

Academic Editor

PLOS ONE